# Cellular and Humoral SARS-CoV-2 Vaccination Responses in 192 Adult Recipients of Allogeneic Hematopoietic Cell Transplantation

**DOI:** 10.3390/vaccines10111782

**Published:** 2022-10-23

**Authors:** Thomas Meyer, Gabriele Ihorst, Ingrid Bartsch, Robert Zeiser, Ralph Wäsch, Hartmut Bertz, Jürgen Finke, Daniela Huzly, Claudia Wehr

**Affiliations:** 1Department of Medicine I/Hematology, Oncology and Stem Cell Transplantation, Faculty of Medicine, Medical Center, University of Freiburg, 79106 Freiburg, Germany; 2Clinical Trials Unit, Faculty of Medicine, Medical Center, University of Freiburg, 79106 Freiburg, Germany; 3Institute of Virology, Faculty of Medicine, University of Freiburg, 79106 Freiburg, Germany

**Keywords:** SARS-CoV-2, allogeneic hematopoietic cell transplantation (allo-HCT), IFN-γ release assay (IGRA), specific antibody titer

## Abstract

To determine factors influencing the vaccination response against SARS-CoV-2 is of importance in recipients of allogeneic hematopoietic cell transplantation (allo-HCT) as they display an increased mortality after SARS-CoV-2 infection, an increased risk of extended viral persistence and reduced vaccination response. Real-life data on anti-SARS-CoV-2-S1-IgG titers (*n* = 192) and IFN-γ release (*n* = 110) of allo-HCT recipients were obtained using commercially available, validated assays after vaccination with either mRNA (Comirnaty™, Pfizer-BioNTech™, NY, US and Mainz, Germany or Spikevax™, Moderna™, Cambridge, Massachusetts, US) or vector-based vaccines (Vaxzevria™,AstraZeneca™, Cambridge, UK or Janssen COVID-19 vaccine™Johnson/Johnson, New Brunswick, New Jersey, US), or after a heterologous protocol (vector/mRNA). Humoral response (78% response rate) was influenced by age, time after transplantation, the usage of antithymocyte globulin (ATG) and ongoing immunosuppression, specifically corticosteroids. High counts of B cells during the vaccination period correlated with a humoral response. Only half (55%) of participants showed a cellular vaccination response. It depended on age, time after transplantation, ongoing immunosuppression with ciclosporin A, chronic graft-versus-host disease (cGvHD) and vaccination type, with vector-based protocols favoring a response. Cellular response failure correlated with a higher CD8+ count and activated/HLA-DR+ T cells one year after transplantation. Our data provide the basis to assess both humoral and cellular responses after SARS-CoV2 vaccination in daily practice, thereby opening up the possibility to identify patients at risk.

## 1. Introduction

Mortality after severe acute respiratory syndrome coronavirus-2 (SARS-CoV-2) infection in recipients of allogeneic hematopoietic cell transplantation (allo-HCT) ranged between 16–40% during the early phase of the pandemic, before the emergence of the omicron variant. Risk factors for SARS-CoV-2 mortality after allo-HCT have been reported to be older age, earlier time after allo-HCT, immunosuppression and male sex [1,2,3]. In healthy individuals, multiple studies have shown that the vaccination against SARS-CoV-2 protects against severe courses of infection [4]. In allo-HCT recipients, studies on SARS-CoV-2 infection and mortality after vaccination are scarce [5]. However, several studies have assessed anti-S1 IgG levels as a surrogate marker for vaccination effectiveness—in most cases, after the use of mRNA vaccines. Risk factors for humoral nonresponders after cellular therapy have slightly varied between cohorts, but B-cell depletion, lymphocyte counts <1000/μL, early vaccination after allo-HCT, ongoing immunosuppression and active graft-versus-host disease (GvHD) have been among the main factors affecting titers [6,7,8,9,10,11,12,13,14,15,16,17]. Although easy to obtain, the humoral immune response only reflects one aspect in adaptive immunity. It has been shown that T cells, especially CD8+, are mobilized early after mRNA vaccination [18] and SARS-CoV-2 infection elicits durable T-cell responses [19]. As there is no standardized routine assay to determine cellular vaccination response, data on T-cell-mediated immunity in allo-HCT recipients after vaccinations are still scarce and results are heterogeneous [9,20,21,22].

Here, we report real-life data on both humoral response and interferon-γ (IFN-γ) release after SARS-CoV-2 vaccination in a large cohort of allo-HCT recipients after different vaccination schemes (mRNA, vector vaccines and heterologous schemes with mRNA/vector). We assessed the impact of transplantation details, lymphocyte subpopulations and the differential effect of single immunosuppressive agents on vaccination responses.

## 2. Methods

### 2.1. Patients

This retrospective single-center study was performed at Freiburg University Medical Center after approval by the local Ethics Committee (EK-FR: 21-1590). We included 192 adult allo-HCT recipients with an assessment of humoral and/or cellular vaccination response on a routine basis and at least one SARS-CoV-2 vaccination. Data were assessed from December 2020 until October 2021.

### 2.2. Detection of SARS-CoV-2 Antibodies

Anti-SARS-CoV-S1-IgG was determined using a commercially available assay detecting IgG antibodies against S1-RBD (SARS-CoV-2 IgG, Siemens™, Munich, Germany). Anti-S1-IgG titers of >21.8 BAU/mL were classified as a humoral response.

### 2.3. SARS-CoV-2 IFN-γ Release Assay

The IFN-γ release assay (IGRA) was performed as described previously with a SARS-CoV2 IGRA (EUROIMMUN™, Lübeck, Germany). In brief, whole blood was incubated in stimulation tubes containing a peptide pool based on the S1 domain of the SARS-CoV-2 spike protein for 20–24 h with positive and negative stimulation controls. IFN-γ concentration was measured via an ELISA [23].

### 2.4. Lymphocyte Counts

Lymphocyte subsets were determined via flow cytometry after full blood staining on an FACS CANTO II™ and analyzed with FACS Diva™ (Becton Dickinson™, Franklin Lakes, New Jersey) at two time points: (1) The peri-vaccination period was defined as 0 to 3 months prior to the last or only documented vaccination date by month. (2) The time period of one year was defined as 330 to 390 days after allo-HCT. If there were multiple measurements during the defined time period, the average value was taken.

### 2.5. Statistical Analysis

Data were analyzed using R-Studio™ (Boston, Massachusetts, US) version 1.4.1717, Apache OpenOffice™ (Wakefield, Massachusetts) version 4.1.7. and Inkscape™ version 1.1.1. For continuous variables, normality was assessed visually. Group comparisons of data with a Gaussian distribution were performed using a two sided *t*-test, and non-normally distributed data were tested using a Wilcoxon rank-sum test. For categorical variables, a chi-squared test was used to assess differences. The associations between various parameters and the humoral and cellular response were analyzed using logistic regression models. The models were determined by including relevant factors based on univariate analysis and performing automated stepwise selection of independent variables using a combination of backward elimination and forward selection.

## 3. Results

### 3.1. Patient Cohort

We included 192 adult allo-HCT recipients (Table 1). Due to the data cut-off date (October 2021), the majority of patients received two vaccinations according to the recommendation of German vaccination guidelines. Patients were vaccinated predominately with mRNA-based vaccines (Comirnaty™, Pfizer-BioNTech™, NY, USA and Mainz, Germany or Spikevax™, Moderna™, Cambridge, MA, USA *n* = 129, 67%). A vector-based immunization (Vaxzevria™,AstraZeneca™, Cambridge, UK or Janssen COVID-19 vaccine™ Johnson/Johnson, New Brunswick, New Jersey,) was used in 10%, *n* = 20. A heterologous protocol of first using a vector-based agent, followed by a booster with an mRNA-based vaccination was used in 10 patients (5%). The heterologous protocol was implemented in Germany in spring 2021 for individuals below the age of 60 after the recognition of vaccine-induced immune thrombotic thrombocytopenia [24]. The time between allo-HCT and best vaccination response ranged between 4 months and 26 years with a median of 3.3 years for humoral and 3.4 years for cellular response. The majority of patients were off immunosuppressive therapy (*n* = 118, 61%) at the time of vaccination. For the 74 individuals (39%) with active immunosuppression, ruxolitinib was the agent most commonly used (*n* = 23), followed by ciclosporin A (*n* = 22) and prednisolone (*n* = 15).

### 3.2. Humoral Vaccination Response Depends on Age, Time after Allo-HCT, Antithymocyte Globulin (ATG) Usage and Active Immunosuppression

Anti-SARS-CoV-S1-IgG was assessed in all 192 patients. Of those, 150 patients (78%) responded, whereas 42 (22%) were nonresponders. Humoral responders were significantly younger compared with nonresponders, with a median age of 60 (IQR = 49.4–67.4) vs. 64.5 years (IQR = 55.3–70.8, *p* < 0.05). Responders were also younger at allo-HCT compared with nonresponders, with a median age at transplantation of 53.5 (IQR = 42.5–62.6) vs. 59.5 (IQR = 50.7–68.9) years (*p* < 0.01, Figure 1A). Additionally, the time from allo-HCT to humoral response was significantly longer in the responder group (median of 4 years, IQR = 1.7–7.7) vs. 1.3 years (IQR = 0.7–4.5, *p* < 0.01) in the nonresponder group (Figure 1A).

GvHD prophylaxis as well as ongoing immunosuppression differed in the two groups (Figure 1B). ATG-based GvHD prophylaxis was more prevalent in nonresponders (Figure 1B, *p* < 0.05). Among the humoral responders, 33% were on active immunosuppression at vaccination compared with 60% of the nonresponders (*p* < 0.05). Particularly, the use of prednisolone (8/15 responders, 53%, *p* < 0.01) and ciclosporin A (14/22 responders, 64%, *p* < 0.05) impaired specific antibody production compared to individuals without active immunosuppression (101/118 responders, 86%). The median daily prednisolone dosage was similar in humoral responders (7.5 mg) and nonresponders (10 mg). Of note, ruxolitinib did not seem to have an impact on humoral response (18/23 responders, 78%) compared to patients off active immunosuppression (86% responders, Figure 1B). Donor type and intensity of conditioning chemotherapy did not influence humoral response rates. Both groups predominantly received reduced toxicity myeloablative conditioning regimens (Figure 1B). Intriguingly, individuals receiving a heterologous vaccination scheme of a one-time vector followed by an mRNA-based vaccine showed a 100% humoral response rate (*n* = 10). However, compared to the response rate of mRNA- and vector-based vaccination of 78% (*n* = 129) and 75% (*n* = 20), respectively, this difference was not statistically significant (Figure 1B).

To consolidate the above results, we conducted a multivariate analysis to predict failure in the humoral response (Appendix A). In concordance to the univariate analysis, the variables determining the model were immunosuppression in general, the usage of prednisolone, the usage of ATG and age at transplantation, confirming the findings of the univariate analysis.

### 3.3. Cellular Vaccination Response Depends on Age, Ciclosporin a Usage, Vaccination Scheme and cGvHD

IFN-γ release after S1 peptide stimulation was determined to assess cellular vaccination response in 110 patients. Of those, 60 individuals (55%) responded, whereas 50 (45%) did not. In healthcare workers, the response rate after vaccination with the same assay has been 100% [23].

Similar to humoral response, in allo-HCT recipients the median age at vaccination (responders 55.5 years, IQR = 45.1–64.6; nonresponder 65.7 years, IQR = 59.7–70.6; *p* < 0.01) and at transplantation (responders 47.6, IQR = 61.0–32.0; nonresponder 62.1 years, IQR = 52.4–66.1; *p* < 0.01; Figure 1A,C) was lower in cellular responders compared with nonresponders. Additionally, the median time from transplantation to response was longer in cellular responders (4.0 years, IQR = 1.6–9.0) compared to nonresponders (median 2.1 years, IQR = 1.0–5.3; *p* = 0.057; Figure 1C).

The use of ATG-based GvHD prophylaxis was equally distributed among cellular responders and nonresponders (63% ATG usage in responders vs. 74% in nonresponders, *p* = 0.32). The influence of active immunosuppressive therapy was not as pronounced for cellular response as for humoral response. Among IGRA responders, 70% (*n* = 42) were off immunosuppressive therapy compared to 56% (*n* = 28) of nonresponders (*p* = 0.19). Among the different immunosuppressive agents, ciclosporin A had a significant impact on cellular response rates (4/14, 29% responders) compared with individuals off immunosuppression (42/70, 60% responders, *p* < 0.05). Patients treated with prednisolone or ruxolitinib had lower but not significantly different IGRA response rates compared to the non-immunosuppressed population (Figure 1D).

Among cellular responders, mismatch unrelated donors (MMUD) were more prevalent compared to nonresponders (MMUD cellular responder 18%, *n* = 11; nonresponder 8%, *n* = 4; *p* = 0.072). Matched related donors (MRD) were less prevalent among cellular responders (MRD cellular responder 17%, *n* = 10; nonresponder 36%, *n* = 18; Figure 1D).

Myeloablative conditioning (MAC) was used in 23% (*n* = 40) of responders and 6% of nonresponders (*n* = 3), possibly reflecting the younger age of MAC recipients. Additionally, reduced toxicity MAC regimens were used less frequently (55%, *n* = 33) in responders versus nonresponders (74%, *n* = 37; *p* < 0.05; Figure 1D).

With regard to the vaccination scheme, patients receiving a one-time vector followed by an mRNA-based vaccine had a 100% cellular response rate (*n* = 8). Patients vaccinated with only vector-based agents showed a response rate of 67% (*n* = 6) and patients immunized with only mRNA-based vaccines, a response rate of 48% (*n* = 75, *p* < 0.05, Figure 1D).

To consolidate the results of the univariate analyses, we used a multivariate analysis to predict cellular response failure. The variables determining the model were the vaccination scheme, age at transplantation, the use of ciclosporin A and if the patient at one time had moderate or severe cGvHD (Appendix A). For the regression model, patients vaccinated with only vector-based protocols and patients receiving the heterologous protocol were grouped due to the low number of patients in each group. The latter showed a significantly smaller likelihood for cellular response failure when compared to only mRNA-based vaccination (OR for nonresponse compared to mRNA 0.12, *p* = 0.01). cGvHD in our univariate analysis did not seem to influence cellular response (univariate 30% of responders vs. 38% of nonresponders with moderate or severe cGvHD, *p* = 0.50). In our bilinear logistic regression model, moderate or severe cGvHD increased the likelihood of cellular response failure upon vaccination (OR for non-response 2.52, *p* = 0.065).

In summary, age, vaccination scheme, cGvHD and use of ciclosporin A influenced IFN-γ release in allo-HCT recipients.

### 3.4. Influence of Lymphocyte Subsets on Humoral and Cellular Vaccination Response

To assess the influence of immunological reconstitution on humoral and cellular immunization responses, we analyzed lymphocyte subpopulations up to three months prior to vaccination (peri-vaccination period *n* = 78 for humoral, *n* = 49 for cellular response assessment, Appendix A). In patients vaccinated more than one year after allo-HCT, we assessed lymphocyte counts in vaccination responders and nonresponders one year after transplantation (Appendix A, *n* = 109 for humoral, *n* = 58 for cellular response assessment).

B-cell counts in the peri-vaccination period were higher in humoral responders compared with nonresponders (median B-cell count 182 cells/µL vs. 28 cells/µL, *p* < 0.05). A trend towards higher B-lymphocyte counts was also observed in cellular responders vs. nonresponders (median B-cell count 180 cells/µL vs. 73 cells/µL, *p* = 0.07). Additionally, peri-vaccine CD4+ T-cell counts were higher in the humoral responder group compared with nonresponders (median CD4+ T-cell count 330/µL vs. 211/µL, *p* = 0.07). In contrast, CD4+-lymphocyte counts were comparable between cellular responders and nonresponders (median CD4+ T-cell count 278/µL vs. 223/µL, *p* = 0.45). NK-cell counts were higher for humoral responders compared to nonresponders (median NK-cell count 196 cells/µL vs. 146 cells/µL, *p* = 0.05). This difference was not as pronounced for cellular responders vs. nonresponders (median NK-cell count 203 cells/µL vs. 166 cells/µL, *p* = 0.42).

At one year after allo-HCT, B-, CD4+ and NK-lymphocyte counts were comparable in humoral responders and nonresponders (median B-cell count: 246 cells/µL vs. 250 cells/µL; CD4+ T cell: 239/µL vs. 196/µL; NK cell: 207 cells/µL vs. 170 cells/µL). Interestingly, cellular responders had lower CD8+-lymphocyte counts at one year after transplantation compared with nonresponders (median CD8+: IGRA responder 364 cells/µL vs. IGRA nonresponder 678 cells/µL, *p* < 0.05). A trend towards higher CD8+-lymphocyte counts in cellular nonresponders was also observed in the peri-vaccination period (median CD8+: IGRA responder 309 cells/µL vs. nonresponder 442 cells/µL, *p* = 0.21). Cellular nonresponders also showed higher activated, HLA-DR+ T-cell counts one year after transplantation compared to responders (median HLA-DR+ T cells IGRA responder 58/µL vs. IGRA nonresponder 88/µL, *p* < 0.05).

In summary, humoral vaccination response is positively associated with B-, CD4+- and NK-cell counts in the peri-vaccination period, whereas higher CD8+-lymphocyte counts seemed to negatively influence IFN-γ release, especially one year after allo-HCT.

## 4. Discussion

In this study, we evaluated the humoral and cellular SARS-CoV-2 vaccination responses of 192 recipients of allo-HCT. Our real-world data show similar humoral response rates of about 80% in accordance to previous studies [16,25,26,27], but were lower compared to the 100% response rate reported in healthy individuals [23,27]. Besides age and time after allo-HCT, antibody titers were reduced upon active glucocorticoid and ciclosporin A usage and ATG-based GvHD prophylaxis, confirming previous reports [10,27]. As our cohort also includes the—to our knowledge—largest cohort of vaccination responses in allo-HCT recipients under ruxolitinib treatment, we were able to investigate the differential effect of JAK inhibition. Interestingly, in contrast to ciclosporin A and corticosteroids, antibody response was not affected by the use of ruxolitinib, thereby pointing towards possible tapering strategies in GvHD treatment. Although it has become evident that sterile immunity against SARS-CoV-2 cannot be achieved, many studies use antibody titers—determined in a variety of assays—as a surrogate marker for protection against SARS-CoV-2 in immunocompromised patients. However, the exact cut-offs for the protection against infection or a severe course of the disease for different variants of the virus have not been determined. Additionally, long-term persistence of B cells without titers has been reported [28]. Thus, humoral response can currently be only an orientation to guide vaccination and infection prevention strategies in immunocompromised persons.

Results of cellular vaccination responses in allo-HCT recipients are heterogeneous and not widely available. The methods used were ELISpot and/or intracellular cytokine staining and reported response rates have ranged from <30% [9,22] to high response rates of >75% [20,21]. We used an easy to perform, standardized and validated assay with a high specificity of 96.3–100% and sensitivity of 75.4–86.9% to detect past infection [23]. This test is easy to implement in routine diagnostics. We observed a response rate of 55% in allo-HCT recipients compared to a 100% response rate in healthcare workers [23], underlining the fact that the cellular immune response in patients after allo-HCT is severely compromised. IFN-γ release was especially reduced with increasing age due to immunosuppression with ciclosporin A and cGvHD. How cellular immunity affects protection against SARS-CoV-2 infection and/or against a severe course, specifically in recipients of allo-HCT, is still to be determined and beyond the scope of this study. However, concerning the preserved effectiveness of cellular response against new variants of SARS-CoV-2 [29], longitudinal studies evaluating protection and infection using the IGRA implemented in routine diagnostics seem feasible.

With regard to lymphocyte subsets, we were able to confirm previous observations, as our cohort humoral response correlated with higher B-, NK- and CD4-cell counts in the vaccination period [9,25,27,30,31]. Intriguingly, cellular response failure in our collective correlated with higher CD8+ and activated HLA-DR+ T cells one year after transplantation. Of note, the median time to best cellular response was 3.4 years in our cohort. Ram and colleagues reported a similar finding with a correlation between a recent low CD4+/CD8+ ratio for cellular response failure [9]. We hypothesize that higher activated CD8-cell counts are surrogate markers for impaired immune reconstitution after allo-HCT. Additionally, an inverted CD4/8 ratio has been reported in the context of immunosenescence in people living with human immunodeficiency virus (HIV) and as a marker of immune aging [32,33].

Due to the changing vaccination recommendations during our data acquisition, our collective contains the largest number of vector-based vaccinated allo-HCT recipients showing both high humoral and cellular response rates. Thus, in the collective of allo-HCT recipients, vector-based vaccination seems to provide superior immunization against SARS-CoV-2. This observation was also reported for the humoral response after the first vaccination [17]. Both mRNA- and vector-based vaccines are applied without adjuvants and elicit an immune response through intrinsic mechanisms. mRNA vaccines use the immune-activating properties of single- and double-stranded RNA via Toll-like receptors (TLR) and the inflammasome. Vector-based vaccines activate the immune system in both a TLR-dependent and -independent way [34,35]. We hypothesize that the broader immune activation with vector-based vaccines might be beneficial in immunocompromised hosts. Further studies will need to investigate if vector-based immunization protocols are more effective in preventing severe SARS-CoV-2 infection in allo-HCT recipients.

In summary, we provide a comprehensive analysis of humoral and cellular vaccination responses in a large cohort of adult allo-HCT recipients. The IGRA used in our manuscript can easily be included into routine practice in the future and provide a broader basis to guide decisions on vaccination strategies in this vulnerable patient cohort.

## Figures and Tables

**Figure 1 vaccines-10-01782-f001:**
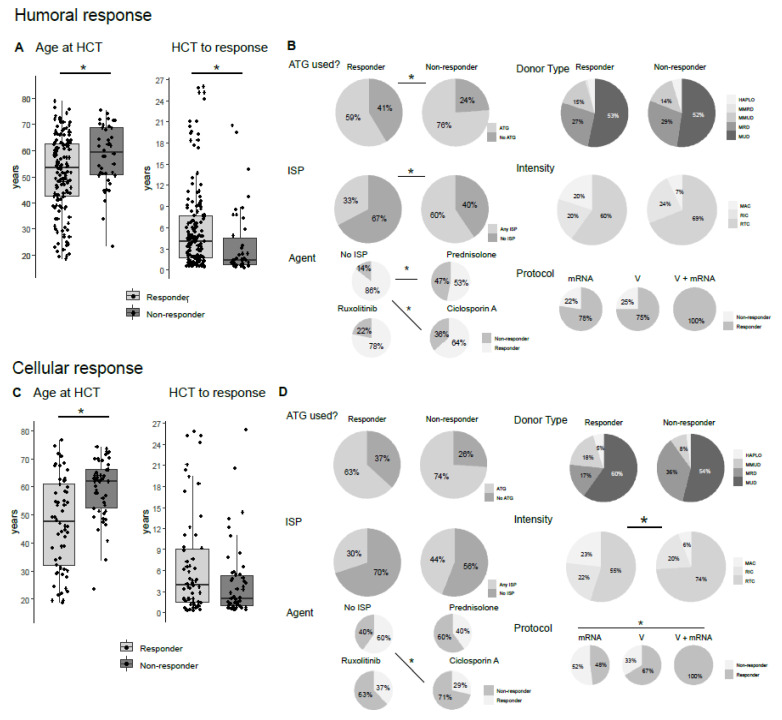
Humoral and cellular SARS-CoV-2 vaccination response. (**A**) Box plots display age at allo-HCT and the time from allo-HCT to best measured humoral (**A**) and cellular response (**C**) stratified according to responders and nonresponders. Points represent individual patients. (**B**,**D**) In the left column, pie charts show frequencies of usage of antithymocyte globulin (ATG), active ongoing treatment with immunosuppressants (ISP) and the usage of specific immunosuppressive agents in responders versus nonresponders. The right column displays frequencies of donor type, intensity of the conditioning chemotherapy and the vaccination protocol used in responders and nonresponders. Matched unrelated donors (MUD), matched related donors (MRD), mismatched unrelated donors (MMUD), mismatched related donors (MMRD), HLA-haploidentical donors (HAPLO), myeloablative conditioning (MAC), reduced toxicity myeloablative conditioning (RTC) and reduced intensity conditioning (RIC). Vaccination protocols displayed are based on only messenger RNA (mRNA) vaccines, only viral vector vaccines (V) or a heterologous combination of one-time viral vector vaccine followed by an mRNA vaccine (V + mRNA). *, *p* < 0.05.

**Table 1 vaccines-10-01782-t001:** Patient characteristics.

Characteristics	Entire Cohort (*n* = 192)
Age at data cut-off (year), median (range)	61 (21–83)
Age at allo-HCT (year), median (range)	54 (19–79)
Time from allo-HCT to best humoral response (years), median (range)	3.3 (0.35–26.03)
Time from allo-HCT to best cellular response (years), median (range)	3.4 (0.41–26.03)
Male, *n* (%)	115 (60)
Female, *n* (%)	77 (40)
Number of vaccinations documented, *n* (%)	
1	192 (100)
2	177 (92)
3	4 (2)
Type of vaccination, *n* (%)	
mRNA	129 (67)
Vector	20 (10)
1 mRNA + 1 Vector	10 (5)
Unconventionally vaccinated	2 (1)
Not documented	31 (16)
Baseline disease, *n* (%)	
AML	78 (41)
ALL	10 (5)
MDS	21 (11)
MDS/MPN	6 (3)
Myeloma	21 (11)
Other NHL	20 (10)
Immunodeficiency	5 (3)
Other	7 (4)
Donor type, *n* (%)	
MUD	102 (53)
MRD	52 (27)
MMUD	29 (15)
MMRD	2 (1)
HAPLO	7 (4)
Graft source, *n* (%)	
PBSC	183 (95)
BM	9 (5)
Conditioning intensity	
MAC	33 (17)
RTC	119 (62)
RIC	40 (21)
CMV, *n* (%)	
−/−	83 (43)
−/+	36 (19)
+/−	23 (12)
+/+	50 (26)
GvHD, *n* (%)	
aGvHD	122 (64)
aGvHD Grade II or more	76 (40)
cGvHD	111 (58)
cGvHD moderate or severe	74 (39)
Active immunosuppression, *n* (%)	
None	118 (61)
Ruxolitinib	23 (12)
Ciclosporin A	22 (11)
Prednisolone	15 (8)
Prednisolone dose (mg), median (range)	10 (2.50–50)
Everolimus	6 (3)
Mycophenolic acid	2 (1)
Other	29 (15)
Combination of 2 agents	11 (6)
Combination of 3 agents	6 (3)
Time from vaccination to response (years)	
For humoral response *n* = 167	
median (IQR)	0.15 (0.07–0.28)
For cellular response *n* = 113	
median (IQR)	0.18 (0.10–0.28)

Legend Table 1: allo-HCT—allogeneic hematopoietic cell transplantation, AML—acute myeloid leukemia, ALL—acute lymphatic leukemia, MDS—myelodysplastic syndrome, MPN—myeloproliferative neoplasia, NHL—non-Hodgkin lymphoma, (M)MUD— (mis)matched unrelated donor, (M)MRD— (mis)matched related donor, HAPLO—haploidentical donor, PBSC—peripheral blood stem cell, BM—bone marrow, MAC—myeloablative, RTC—reduced toxicity MAC, RIC—reduced intensity, CMV—cytomegalovirus, (a/c)GvHD—(acute/chronic) graft-versus-host disease.

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
