# Peer review of "Cellular and Humoral SARS-CoV-2 Vaccination Responses in 192 Adult Recipients of Allogeneic Hematopoietic Cell Transplantation"

_vaccines, 2022, doi:10.3390/vaccines10111782_

Round 1

Reviewer 1 Report

Authors studied cellular  and humoral SARS-CoV2 vaccination responses in adult recipients of allogenic hematopoietic cell transplantation. Results are clear and new.

Minor comment:

Authors had better discuss CD8+ CTL activity of SARS-CoV2 -vaccinated patients receiving allogenic hematopoietic cell transplantation.

Author Response

We thank the reviewer for concise review: In our study we observed a correlation of increased activated CD8+ T lymphocyte counts with cellular response failure after vaccination. We hypothesize that the increased activated CD8 cell counts are a surrogate marker for failed immunological reconstitution and immunosenescence (cf. discussion line 279-289). Unfortunately, we do not have data on IFN-gamma release of CD8+ T cells vs CD4+ T cells vs NK cells as our IGRA uses whole blood as starting material.  We therefore did not comment on the activity of CD8+ T lymphocytes in allo-HCT recipients. 

Reviewer 2 Report

Meyer and collaborators' work tries to demonstrate the relationship between the cellular and humoral response post-vaccination COVID-19 in patients with hematopoietic cell transplantation. The evidence obtained is clear and relevant to consider the effect of immunosuppressant treatments and cell transplantation on the cellular and humoral response against infection. Furthermore, their findings are of great relevance for the knowledge of the impact of vaccination in this type of population, so I consider its publication in the journal necessary.

Author Response

We thank for fast and concise review of our data.

Reviewer 3 Report

The Manuscript entitled "Cellular and humoral SARS-CoV2 vaccination responses in 192 adult recipients of allogeneic hematopoietic cell transplantation” is an interesting manuscript about the humoral and cellular immune response against SARS-CoV2 vaccination in a large cohort of patients recipients of allogeneic hematopoietic cell transplantation. The study is well designed, well written, concise and will contribute to the Immunology and Infectious Diseases fields. Furthermore, I couldn’t detect any flaw in the manuscript and in this reviewer opinion, the manuscript should be accepted for publication in Vaccines after minor revision.

Specific comments:

1- Title: Please, fix the typo in “recipientsa”. 

2- Abstract: Please, describe the full names of the “ATG” and “GvHD” abbreviations as they first appears in the text. 

3- Table 1: It is important to include the full names of all abbreviations in the legend of the Table.

Author Response

We thank the reviewer for valuable comments. All suggestions were incorporated into the manuscript.